# Genome-Wide Identification and Characterization of the VQ Motif-Containing Gene Family Based on Their Evolution and Expression Analysis under Abiotic Stress and Hormone Treatments in Foxtail Millet (*Setaria italica* L.)

**DOI:** 10.3390/genes14051032

**Published:** 2023-04-30

**Authors:** Meiling Liu, Cong Li, Yuntong Li, Yingtai An, Xiaoxi Ruan, Yicheng Guo, Xiaomei Dong, Yanye Ruan

**Affiliations:** 1College of Bioscience and Biotechnology, Shenyang Agricultural University, Shenyang 110866, China; 2School of Materials Science and Engineering, Northeastern University, Shenyang 110819, China; 3Shenyang Key Laboratory of Maize Genomic Selection Breeding, Shenyang Agricultural University, Shenyang 110866, China

**Keywords:** foxtail millet, *VQ* gene family, expression patterns, plant hormone, abiotic stress

## Abstract

Valine–glutamine (VQ) motif-containing proteins are transcriptional regulatory cofactors that play critical roles in plant growth and response to biotic and abiotic stresses. However, information on the *VQ* gene family in foxtail millet (*Setaria italica* L.) is currently limited. In this study, a total of 32 *SiVQ* genes were identified in foxtail millet and classified into seven groups (I–VII), based on the constructed phylogenetic relationships; the protein-conserved motif showed high similarity within each group. Gene structure analysis showed that most *SiVQs* had no introns. Whole-genome duplication analysis revealed that segmental duplications contributed to the expansion of the *SiVQ* gene family. The cis-element analysis demonstrated that growth and development, stress response, and hormone-response-related cis-elements were all widely distributed in the promoters of the *SiVQs*. Gene expression analysis demonstrated that the expression of most *SiVQ* genes was induced by abiotic stress and phytohormone treatments, and seven *SiVQ* genes showed significant upregulation under both abiotic stress and phytohormone treatments. A potential interaction network between SiVQs and SiWRKYs was predicted. This research provides a basis to further investigate the molecular function of *VQs* in plant growth and abiotic stress responses.

## 1. Introduction

Valine–glutamine (VQ) motif-containing proteins are transcriptional regulators that interact with known transcription factors (TFs), such as WRKY, PIF, and bHLH-type TFs, that are involved in regulating plant growth [1,2]. This interaction depends on the highly conserved VQ motif (FxxhVQxhTG) in VQ proteins [3]. Among these amino acid residues, x is any amino acid, h is a hydrophobic residue, and the last three residues show a variety of differences; depending on those differences, the VQ motif can be divided into different types, namely LTG, FTG, VTG, YTG, ITG, ATG, LTS, LTD, LTR, LTV, and LTA [3]. Aside from the highly conserved VQ motif, the amino acid sequences at other positions of VQ proteins show differences, leading to functional diversity among VQ family members [4].

VQ proteins are involved in seed germination [5], seedling development [2], leaf senescence [6], flowering [7], and pollen development [8]. In Arabidopsis, mutations in *AtVQ20*, *AtWRKY2*, and *AtWRKY34* simultaneously cause male sterility and defects in pollen development. Further studies have shown that the AtVQ20 protein interacts with the TFs AtWRKY2 and AtWRKY34, affecting their transcriptional functions and subsequently regulating male gametogenesis [8]. In rice, the knockout of *OsVQ1* can suppress the expression of the genes that promote flowering, resulting in late flowering in mutant plants under long-day conditions [7]. In tomato plants, the SlVQ7 protein interacts with SlWRKY37 to increase the transcriptional activation ability of SlWRKY37, thereby promoting the expression of the target genes *SlSGR1* and *SlWRKY53* and accelerating dark- and JA-induced leaf senescence [6]. VQs are also involved in the response to biotic stresses such as pathogens [9,10], and abiotic stresses such as high temperature [11,12], salt [13], and drought [14]. OsVQ25 interacts with the TFs OsPUB73 and OsWRKY53 to balance plant broad-spectrum disease resistance and growth in rice [15]. In apple, MdVQ37 regulates the expression of the genes involved in SA catabolism. Under high-temperature stress, endogenous SA levels are reduced, and SA-dependent signaling pathways are disrupted, resulting in heat sensitivity in overexpressing *MdVQ37* plants [11]. VQs are also involved in plant hormone signaling, including responses to ABA [16], SA [11], and JA [17,18]. Thus, VQ proteins can influence the transcriptional activity of TFs and are involved in plant development, stress responses, and hormone signaling pathways [19]. Therefore, mining the potential *VQ* genes in plants will provide a useful basis for the analysis of plant development and stress response mechanisms at the post-translational level.

Currently, 34, 39, 61, 65, and 74 *VQs* have been identified in Arabidopsis [3], rice [20], maize [21], wheat [22], and soybean [23], respectively. Conserved motif analysis revealed that six and four VQ motif types were found in Arabidopsis and rice, respectively, and phylogenetic analysis enabled Arabidopsis and rice VQ proteins to be divided into seven groups (groups I–VII), among which groups V and VI are FTG and VTG/ITG types, respectively, and the remaining groups are the LTG type. Maize has six VQ motif types, and the maize VQ proteins are classified into nine groups (groups I–IX). Among those, maize group I belongs to the FTG/VTG/ATG/ITG types, which corresponds to groups V and VI of Arabidopsis and rice, while groups II to IX in maize belong to the LTG type. The results of gene structure analysis have revealed that most *VQs* lack introns. The biological functions of *VQs* have been explored using quantitative real-time PCR (qRT-PCR) and RNA-seq data, showing that the expression of many *VQs* significantly changes under stress or hormone treatments.

Foxtail millet is a traditional grain crop in China and has advantages for research due to its small genome size, strong abiotic stress resistance, and short growth period [24,25]. The potential tolerance to abiotic stress in foxtail millet has rendered it a favorable research subject for studying the molecular mechanisms of abiotic stress resistance [26]. Thus, it is emerging as a new model plant of Gramineae. Although the important biological role of VQs has been demonstrated, little information is currently available on the *VQ* gene family in foxtail millet, and how those genes might be involved in plant growth and response to biotic and abiotic stresses.

In the present study, we identified 32 putative *SiVQs* in foxtail millet and carried out comprehensive bioinformatic analyses, including analysis of gene location and structure, phylogenetic relationships, synteny, and cis-elements. We also performed the expression analysis of *SiVQ* genes under abiotic stresses and phytohormone treatments and predicted the interaction network between SiVQs and SiWRKYs. Our results provide fundamental information for the characterization and evolution of *SiVQ* genes and help build our understanding of the functions of *SiVQs*, which will be valuable for further research on the biological function of *SiVQs* involved in plant growth and stress responses.

## 2. Materials and Methods

### 2.1. Identification, Chromosomal Localization, and Phylogenetic Analysis of SiVQs

In the present study, the foxtail millet genome sequence was downloaded from the Ensembl Plant database (http://plants.ensembl.org/index.html (accessed on 18 April 2022)). The hidden Markov model (HMM) profile of the VQ domain (PF05678) was obtained from the Pfam database (http://pfam.xfam.org/ (accessed on 18 April 2022)) [27]. The *SiVQ* family members were retrieved from the foxtail millet genome database using the HMMSEARCH program of the HMMER software. The online program SMART tool (http://smart.embl-heidelberg.de/ (accessed on 20 April 2022)) and the Conserved Domains Database Search (https://www.ncbi.nlm.nih.gov/Structure/cdd/wrpsb.cgi (accessed on 20 April 2022)) were used to ultimately determine the *SiVQ* genes. The biophysical properties of the SiVQs, including peptide length, isoelectric point (pI), and molecular weight (MW), were estimated using the online program ExPasy (http://www.expasy.org/tools/ (accessed on 23 April 2022)) [28]. The physical locations of the *SiVQs* on the chromosomes were visualized using the Mapchart software [29], and the subcellular localization prediction tool WoLF PSORT (https://www.genscript.com/wolf-psort.html (accessed on 24 April 2022)) was used to predict the likely location of the SiVQs.

According to previous research on the genome-wide identification of AtVQ and OsVQ proteins in Arabidopsis and rice [3,20], 34 Arabidopsis and 36 rice VQ protein sequences were downloaded from the Phytozome database (https://phytozome-next.jgi.doe.gov/ (accessed on 26 April 2022)). Based on the AtVQ and OsVQ protein sequences, combined with all the SiVQs, a multiple sequence alignment was carried out using the ClustalW tool to study the evolutionary relationships and classification of the SiVQs. Depending on the alignment results, the phylogenetic tree was constructed using the MEGA11 software with the maximum likelihood method, and the parameters were as follows: JTT matrix-based model, complete deletion, and 1000 bootstrap replications [30]. The online tool Evolview (https://evolgenius.info//evolview-v2/#login (accessed on 29 April 2022)) was used to draw the phylogenetic tree.

### 2.2. Conserved Sequence and Gene Structure Analysis

The conserved motifs of the SiVQs were detected using the online program multiple expectation maximization for motif elicitation (MEME) (http://meme.ebi.edu.au/ (accessed on 1 May 2022)) with the following parameters: The maximum number of motifs was five, and the optimal width ranged from 6 to 100 [31]. Conserved motifs were drawn using the TBtools software. The exon and intron structures were determined by aligning the coding sequences (CDSs) of the *SiVQ* genes with their corresponding genomic DNA sequences. The gene structures of the *SiVQs* were visualized using the online program Gene Structure Display Server (GSDS) (http://gsds.cbi.puk.edu.cn/ (accessed on 3 May 2022)).

### 2.3. Gene Duplication and Collinearity Analysis

To identify segmental and tandem duplication events in the foxtail millet *SiVQ* gene family, the amino acid sequences of all SiVQs were aligned using BLASTp at a 1 × 10^−10^ significance level. As described in previous research, two or more *SiVQ* genes located within 200 kb of each other on the same chromosome and sharing more than 70% identity can be defined as tandem duplication events [32]. In addition, we used a multiple collinear scanning kit (MCScanX) to identify segmental duplication events [33]. The collinearity maps were drawn using the Circos program to visualize the duplicated gene pairs between the *SiVQ* genes, and the orthologous *VQ* gene pairs between foxtail millet and Arabidopsis (*Arabidopsis thaliana*), tomato (*Solanum lycopersicum*), alfalfa (*Medicago truncatula* ), rice (*Oryza sativa*), sorghum (*Sorghum bicolor*), and maize (*Zea mays*), respectively. The values of nonsynonymous (Ka) and synonymous (Ks) substitutions of duplicated *VQ* gene pairs were calculated to evaluate the divergence time and selection history using the KaKs_Calculator 2.0 [34]. The divergence time (T) for each duplicated gene pair was estimated using the following equation: T = Ks/(2 × 6.5 × 10^−9^) × 10^−6^ million years ago (MYA) [35].

### 2.4. Cis-Element Analysis of SiVQ Genes in Foxtail Millet

The upstream regulatory regions of the *SiVQ* genes (the genomic DNA sequences 2000 bp upstream of the cDNA) were retrieved from the foxtail millet genome and were used to predict the potential cis-acting element using the PlantCARE website (https://bioinformatics.psb.ugent.be/webtools/plantcare/html/ (accessed on 7 May 2022)) [36]. The results were categorized, counted, and ultimately visualized using the TBtools software [37].

### 2.5. Analysis of Gene Expression Patterns and Potential Protein Interactions in SiVQs

The transcriptome data of *SiVQs* in 11 different tissues were obtained from the NGDC (https://ngdc.cncb.ac.cn/ (accessed on 11 May 2022)), with accession number PRJCA001726. Transcript abundance was described as fragments per kilobase of exon per million mapped fragments (FPKMs) [38]. If the FPKM value was less than 1 in all tissues, the gene was considered to be non-expressed. If the expression of a gene in one tissue was 10-fold higher than the sum of the expression of this gene in other tissues, this gene was considered to be tissue-specifically expressed, and if the expression of a gene in most tissues was 5-fold higher than other genes, this gene was considered to be highly expressed. The log10-transformed FPKM values were used to draw heatmaps of the expression profiles using the TBtools software. The prediction of interaction networks between SiVQs and SiWRKYs was achieved using the STRING online database with the confidence parameter set to a threshold of 0.4 and the other parameters were as default, and the Cytoscape 3.9.1 software was used to draw the interaction network [39].

### 2.6. Plant Materials and Treatments

Foxtail millet Yugu1 was used in the present study. Seeds were sown evenly in seedling pots covered with vermiculite and watered with distilled water for seed germination. Seedlings were cultured in a greenhouse kept at 24 °C with an 18/6 h photoperiod (day/night) using Hoagland’s nutrient solution (pH 6.0) [40]. Three-leaf stage seedlings were treated with three different abiotic stress treatments: 20% PEG6000 (drought), 200 mM NaCl (salt), 4 °C (cold), and four different hormone treatments: 200 µM abscisic acid (ABA), 0.1 mM salicylic acid (SA), 0.1 mM gibberellin (GA), and 100 µM methyl jasmone (MeJA) [41,42,43]. Plant material was sampled at 0, 6, 12, and 24 h after treatment with three biological replicates for each treatment. All the samples were frozen in liquid nitrogen and stored at −80 °C for subsequent total RNA extraction.

### 2.7. RNA Extraction and Quantitative Real-Time PCR (qRT-PCR)

A plant RNA extraction kit (Accurate Biotechnology Co., Ltd., Changsha, China) was used to extract the total RNA from foxtail millet tissue, according to the manufacturer’s instructions. The M-MLV reverse transcriptase (Promega, Shanghai, China) was used to synthesize the cDNA using RNA as a template. qRT-PCR was carried out using a GoTaq^®^ qPCR Master Mix kit (Promega) and the C1000 real-time PCR system (Bio-Rad, Hercules, CA, USA). All reactions were conducted with three independent biological replicates. The relative expression level was calculated using the 2^−ΔΔCt^ method and compared with that of the control sample at 0 h [44]. The *SETIT_010361mg* was used as an internal control [41]. The primer sequences used in the present study are listed in Appendix A. Significant differences between the treatment and control values were tested through one-way ANOVA analysis at a significance level of *p* = 0.05 using the IBM SPSS Statistics software.

## 3. Results

### 3.1. Identification of VQ Gene Family Members in Foxtail Millet

In total, 32 *VQs* were identified in the foxtail millet, designated *SiVQ01* to *SiVQ32* based on their chromosomal physical location (Table 1). These 32 *SiVQs* were not evenly distributed across all 9 chromosomes (Figure 1). In particular, chromosome IX had the maximum number of seven *SiVQs*, while no *SiVQ* gene was detected on chromosome VIII. Meanwhile, based on the sequence alignment results of the SiVQs, no tandem duplication event was detected in the millet genome.

The detailed data for each *SiVQ*, including gene ID, genomic position, VQ protein length, VQ motif type, isoelectric point, molecular weight, and subcellular localization, are shown in Table 1. The nucleotide lengths of these *SiVQs* ranged from 257 bp (*SiVQ09*) to 1970 bp (*SiVQ14*), and the length of their encoded protein sequences ranged from 85 amino acids (SiVQ09 and SiVQ18) to 390 amino acids (SiVQ22). The analysis of physiochemical properties further revealed that SiVQ proteins widely varied in molecular weight (MW), ranging from 8.95 (SiVQ09) to 38.97 kDa (SiVQ22). The isoelectric point (pI) of the SiVQs varied from 5.08 to 11.21, with a minimum pI noted for SiVQ10 and a maximum pI noted for SiVQ25 (Table 1). In addition, four VQ motif types (LTG, FTG, ITG, and VTG) were identified. Most SiVQs belonged to the LTG type (26/32), four SiVQ proteins belonged to the FTG type, one SiVQ protein belonged to the ITG type, and one SiVQ protein belonged to the VTG type (Table 1). The subcellular localization analysis revealed that 18 SiVQs were predicted as nuclear-localized proteins, 10 were predicted as chloroplast proteins, and 4 were predicted as mitochondrial proteins.

### 3.2. Phylogenetic, Conserved Motif, and Gene Structure Analysis of SiVQs

Notably, 102 VQ protein sequences, including 34 Arabidopsis AtVQs, 36 rice OsVQs, and 32 foxtail millet SiVQs, were used for the phylogenetic analysis to explore the evolutionary relationship of the SiVQs (Appendix A and Figure 2). Based on the classification of Arabidopsis and rice VQs [20], the SiVQs were classified into seven groups (I–VII). Groups VI and VII contained the least number of SiVQs with two members, group II had the largest number of SiVQs with ten members, and groups I, III, IV, and V had six, four, four, and four SiVQs, respectively. According to the results of the multiple sequence alignment of SiVQ proteins, group V belonged to the FTG type, group VI belonged to VTG and ITG types, and the remaining groups belonged to the LTG type (Figure 3), which is consistent with the results found for Arabidopsis and rice [20].

The conserved motifs analysis of the SiVQs revealed that all SiVQs had motif 1, which corresponds to the VQ motif (Figure 4A). It is noteworthy that SiVQs within the same group had similar conserved motifs, whereas SiVQs had differences in motifs between groups, also indicating that the phylogenetic classification was reliable. All members of groups I, III, V, VI, and VII contained only motif 1 except SiVQ16 and SiVQ32. Group IV contained motifs 2, 3, and 4, most members of group II contained motif 5, and SiVQ23 and SiVQ03 also contained motif 4 (Figure 4A). The exon–intron structure of all 32 *SiVQ* genes was analyzed (Figure 4B). Most of them (27 *SiVQs*) had no intron, and only 5 *SiVQs* had 1 intron.

### 3.3. Synteny Analysis of SiVQ Genes

To understand the evolutionary mechanism of *SiVQ* in foxtail millet, the segmental duplication events of the *SiVQ* gene family were analyzed. Following the BLAST results, a total of three homologous gene pairs with six *SiVQs* were identified in the whole genome (Figure 5). Group II exhibited two segmental duplication events, whereas group IV exhibited one segmental duplication event. The selective evolutionary pressure on all three homologous *SiVQ* gene pairs was investigated by calculating the *K_a_*, *K*_s_, and *K*_a_/*K*_s_ ratios of the duplication events. The *K_a_*/*K_s_* values of two of the gene pairs were more than 1.0 (Table 2). In addition, the divergence times were calculated to estimate the SiVQ duplication events of the syntenic blocks (Table 2), and the divergence time of the three *SiVQ*-duplicated gene pairs varied from 24.0 to 45.7 MYA.

The synteny analysis was also conducted between foxtail millet and six other species: three dicot species, namely Arabidopsis, alfalfa, and tomato (*Arabidopsis thaliana*, *Medicago truncatula*, and *Solanum lycopersicum*), and three monocot species, i.e., maize, rice, and sorghum (*Zea mays*, *Oryza sativa*, and *Sorghum bicolor*). Overall, the *SiVQs* had the most orthologous gene pairs with maize (*n* = 41), followed by sorghum (*n* = 29), and rice (*n* = 22). Only two orthologous gene pairs were observed between foxtail millet and Arabidopsis, but no orthologous gene pairs were observed between foxtail millet and tomato or alfalfa, respectively (Figure 6). In addition, 19 *SiVQs* had a collinear relationship with the *VQs* of all the three monocots studied (Appendix A), indicating that these genes are evolutionarily conserved and may play a vital role in the evolution of the *VQ* gene family. The divergence times were also calculated to estimate the *VQ* duplication events between foxtail millet and the three monocots, and the divergence times of these *VQ* gene pairs ranged from 8.3 to 141.6 MYA (Appendix A).

### 3.4. Cis-Element Analysis of the SiVQ Gene Promoter Region

Cis-acting elements in the gene promoter regions were identified to explore the regulatory mechanism of *SiVQs*. In total, 105 cis-elements were obtained and further classified into 5 categories: phytohormone responsive, abiotic and biotic stress, plant growth and development, light responsive, and unknown functions (Appendix A and Figure 7).

In the phytohormone-responsive category, 17 cis-elements associated with different phytohormone responses such as ABA, auxin, ethylene, GA, SA, and MeJA were observed. The largest proportion of these elements was the ABRE element involved in responding to ABA, which represented 22.7%. The second majority was the as-1 element involved in responding to SA, and the TGACG and CGTCA motifs involved in responding to MeJA, representing 16.0% of the total found in this category (Appendix A and Figure 7).

Regarding the abiotic and biotic stress category, the cis-acting elements responsive to drought, salt, and temperature, such as MYB, MBS, STRE, MYC, CCAAT box, the CCGTCC motif, the DRE core, and LTR motifs, represented 78% of the total elements. Additionally, other stress-specific cis-elements were also identified, such as the W box, the TC-rich repeat, the box S, and the WUN motif in response to the wounding and pathogens, and ARE and GC motifs in response to anaerobic conditions (Appendix A and Figure 7).

A total of 27 elements belonged to the light-responsive elements. Most of them were the G box, representing 32.0% of the total identified in this category. Other specific elements were also found, such as circadian (Appendix A). In the plant growth and development category, 13 cis-elements were observed, the majority of which belonged to the CAAT-box motif involved in transcriptional regulation, representing 87.4% of the cis-acting elements present within this category. Several elements associated with meristem development were identified, such as CAT box, CCGTCC box, dOCT, NON box, and O2 site. Some elements associated with endosperm and seed development, such as the GCN4 motif and the RY element, were also observed (Figure 7 and Appendix A).

### 3.5. Expression Patterns of SiVQ Genes in Different Tissues

To characterize the expression patterns of *SiVQs*, the expression levels of the 32 *SiVQs* were analyzed in 11 different tissues to gain a preliminary insight into their potential functions (Figure 8). Based on the FPKM values, we found that 11 *SiVQ* genes (*SiVQ03*, *SiVQ11*, *SiVQ12*, *SiVQ13*, *SiVQ14*, *SiVQ16*, *SiVQ19*, *SiVQ20*, *SiVQ23*, *SiVQ27*, and *SiVQ32*) were highly expressed in different tissues, which were widely distributed in groups I to V, most of which belonged to groups II and III. By contrast, six *SiVQ* genes (*SiVQ04*, *SiVQ07*, *SiVQ09*, *SiVQ10*, *SiVQ21*, and *SiVQ30*) were low or not expressed in different tissues, which were clustered in groups I and II. (Figure 8). Certain genes had high expression levels in only one tissue. For example, *SiVQ15* was highly expressed only in the root. Interestingly, the two *SiVQs* (*SiVQ28* and *SiVQ29*) clustered in group VII showed similar expression and were relatively strongly expressed in the seed, and the paralogous gene pairs *SiVQ03* and *SiVQ23*, as well as *SiVQ11* and *SiVQ19*, also showed very similar expression patterns, with highest levels in the root. These results implicate that these *SiVQs* may have different roles during plant growth and development and that *SiVQs* within the same group may have similar or redundant functions.

### 3.6. Expression Analysis of SiVQ Genes under Abiotic Stress and Exogenous Hormone Treatments

Many *VQs* have been implicated to be involved in abiotic stresses [11,12]. To clarify whether *SiVQ* genes respond to abiotic stresses, the expression levels of *SiVQs* under drought, salt, and low temperature were analyzed using qRT-PCR. Overall, many *SiVQs* exhibited upregulated expression under abiotic stress conditions (Figure 9). Under drought treatment, all *SiVQs* except SiVQ32 showed upregulated expression levels. Six and eight *SiVQ* genes started to respond to drought stress at 6 h and 12 h, respectively. Additionally, most of the upregulated *SiVQ* genes peaked at 12 h. Notably, the expression of four genes (*SiVQ10*, *SiVQ18*, *SiVQ26*, and *SiVQ28*) increased at least 10-fold under drought treatment (Figure 9A and Appendix A). Following salt stress treatment, all *SiVQs* exhibited upregulated expression levels. The expression of four *SiVQ* genes (*SiVQ06*, *SiVQ07*, *SiVQ08*, and *SiVQ10*) continuously increased and peaked at 24 h, while the expression of other *SiVQs* first increased and then decreased. In particular, the expression levels of six *SiVQs* (*SiVQ05*, *SiVQ12*, *SiVQ18*, *SiVQ19*, *SiVQ22*, and *SiVQ23*) were significantly upregulated (>20 fold) under salt stress (Figure 9B and Appendix A). Under cold treatment, the expression levels of 22 genes were induced, and the expression levels of 5 genes were inhibited. Overall, the response to cold was not as significant as the response to drought and salt stress for all *SiVQs*. Only four genes (*SiVQ06*, *SiVQ14*, *SiVQ24*, and *SiVQ25*) showed the most significant upregulated expression (>8 fold), and one gene (*SiVQ07*) showed significantly downregulated expression (13-fold) (Figure 9C and Appendix A).

Furthermore, changes in the expression levels of *SiVQs* were analyzed under four exogenous phytohormone treatments (MeJA, SA, GA, and ABA) (Figure 10). All SiVQ genes exhibited upregulated expression, peaking at 6 h or 12 h after GA treatment. Seven *SiVQ* genes (*SiVQ13*, *SiVQ17*, *SiVQ19*, *SiVQ20*, *SiVQ22*, *SiVQ25*, and *SiVQ28*) responded significantly to GA treatment, with >12-fold changes (Figure 10A and Appendix A). Under MeJA treatment, the expression of most *SiVQ* genes was upregulated at early treatment time points, and five *SiVQ* genes (*SiVQ12*, *SiVQ17*, *SiVQ18*, *SiVQ23*, and *SiVQ27*) responded significantly to MeJA treatment, with >20-fold changes (Figure 10B and Appendix A). The expression levels of *SiVQ* genes were also analyzed in seedlings under SA and ABA treatments. Eight *SiVQ* genes (*SiVQ05*, *SiVQ08*, *SiVQ13*, *SiVQ15*, *SiVQ18*, *SiVQ22*, *SiVQ24*, and *SiVQ31*) and three *SiVQ* genes (*SiVQ12*, *SiVQ25*, and *SiVQ28*) responded significantly (>20-fold) to SA and ABA treatments, respectively (Figure 10C,D and Appendix A).

### 3.7. The Interaction Network of SiVQs with SiWRKYs

A protein interaction network between SiVQs and SiWRKYs was constructed using the STRING database to better understand the potential interactions. Notably, 15 SiVQs were predicted to interact with 28 SiWRKYs (Figure 11 and Appendix A). Among these 15 SiVQ proteins, 3 SiVQs (SiVQ11, SiVQ17, and SiVQ19) interacted with only 1 SiWRKY, while 12 SiVQs interacted with more than 1 SiWRKY. Five SiVQs (SiVQ03, SiVQ20, SiVQ21, SiVQ23, and SiVQ32) interacted with more than seven SiWRKYs to generate a key node (Figure 11). Similar to some WRKYs [23,45], most SiVQ proteins interacted with other SiVQ proteins to function. These results may provide helpful information for future studies on the regulatory network of VQs.

## 4. Discussion

Previous studies have revealed that VQ proteins play an important role in plant growth and stress responses [46,47]. The *VQ* gene family has been systematically analyzed in many species, including cucumber [48], sugarcane [49], and tea [50], and the genes have been proven to respond to various abiotic and biotic stresses. However, limited information on VQ characteristics in foxtail millet is available. Thus, a comprehensive bioinformatic analysis of foxtail millet *VQs* and their expression analysis under different abiotic stresses and hormonal treatments can provide a basis for functional studies of *SiVQs*.

In this study, we identified 32 *SiVQ* genes in foxtail millet, a similar number to that in rice and Arabidopsis but significantly less than in maize, wheat, and soybean, indicating that the number of *VQs* in the genome might be largely dependent on the evolutionary position and/or genome size of the species. The SiVQs were classified into seven groups from our phylogenetic analysis, and we found these SiVQs had a close affinity to the VQs of the monocot rice but a relatively low distant affinity to the VQs of the dicot Arabidopsis. This suggests that SiVQs are conserved in the evolutionary history of monocots. Additionally, the SiVQs in the same group shared similar conserved motifs, suggesting a potential functional similarity between SiVQs within the same group. Most *SiVQs* lacked introns, which is consistent with previous research on VQs in other species, including Arabidopsis [3], rice [20], maize [21], tomato [51], and apple [52]. This might be the result of the loss of introns in plant *VQ* genes during their evolutionary history [48].

Segmental duplication is a major mechanism for the amplification of the *VQ* gene family [53]. We found three segmental duplication events among the *SiVQ* genes, while no tandem duplication was detected, indicating that segmental duplication was a major mode of expansion of the *SiVQ* gene family. The Ka/Ks ratio was calculated to investigate whether the *SiVQ* genes underwent selection pressure. The results demonstrate that positive selection (Ka/Ks > 1) and purifying selection (Ka/Ks < 1) play a role in the evolution of the *SiVQ* gene family. Furthermore, many orthologous gene pairs were detected between foxtail millet and monocot species, whereas few gene pairs were observed between foxtail millet and dicot species. The results of synteny analysis further indicate that the *VQ* genes are highly conserved in monocots. We calculated the divergence time to estimate the VQ duplication events between foxtail millet and three monocot species. Previous studies have demonstrated that foxtail millet diverged from sorghum and maize before ~27 MYA and from rice before ~48 MYA [54]. In this study, we found that the divergence time of most VQ duplication events (61/69) between foxtail millet and maize or sorghum was greater than 27 MYA, and the divergence time of most VQ duplication events (17/22) between foxtail millet and rice was also greater than 48 MYA. These results suggest that the genome duplication of the *VQ* genes in foxtail millet mainly occurred before the divergence of the Gramineae and after the divergence of dicots and monocots.

In this study, many cis-acting elements associated with responses to drought, salt, and cold stresses were found in the promoter region of *SiVQs*, suggesting that they might play a role in abiotic stress responses. Therefore, the expression levels of *SiVQs* under these three abiotic stress treatments were further analyzed. The expression of *SiVQ10*, *SiVQ18*, *SiVQ26*, and *SiVQ28* showed significant upregulation under drought treatment, and drought-stress-responsive cis-acting elements were found in their promoters. Among them, *SiVQ10* and *SiVQ18* belonged to group I, the same group as *OsVQ02* in rice, and *SiVQ18* was orthologous to *OsVQ02*. *SiVQ26* and *SiVQ28* were in group II and group VII, the same groups as *OsVQ14* and *OsVQ36* in rice, respectively. The rice genes *OsVQ02*, *OsVQ14*, and *OsVQ36* were all upregulated under drought treatment [20]. We, therefore, speculate that *SiVQ10*, *SiVQ18*, *SiVQ26*, and *SiVQ28* might be involved in responding to drought stress in foxtail millet. *SiVQ06*, *SiVQ14*, *SiVQ24*, and *SiVQ25* exhibited significant upregulation under cold treatment, and cis-acting elements associated with responses to low temperature were detected in their promoters. The fold change in expression was largest for *SiVQ06* and *SiVQ25*, both of which belong to group VI and are probably functionally similar. Previous studies in Arabidopsis found that the expression levels of *AtVQ15* and *AtVQ24*, which belong to group VI, were also significantly upregulated under low-temperature treatment [55]. We, therefore, speculate that *SiVQ06* and *SiVQ25* might be involved in response to cold stress in foxtail millet. *SiVQ05*, *SiVQ12*, *SiVQ18*, *SiVQ19*, *SiVQ22*, and *SiVQ23* showed significant upregulation under salt treatment, and salt-stress-responsive cis-acting elements were found in their promoters. The fold change in expression patterns was largest for *SiVQ12*, *SiVQ18*, and *SiVQ19*, belonging to groups I, II, and IV, respectively. Previous studies found that most VQs belonging to groups I, II, and IV respond to salt stress in Arabidopsis [55]. Therefore, we speculate that *SiVQ12*, *SiVQ18*, and *SiVQ19* might be implicated in the salt stress response in foxtail millet. However, the *SiVQ*-duplicated genes in foxtail millet showed diverse expression profiles. It seems that mutations in critical regions, including promoters and coding sites, might have altered their gene expression patterns.

Previous studies have provided evidence indicating that plant hormones can regulate the expression of *VQs* [49]. The cis-acting elements associated with responses to various hormones were found in the promoters of *SiVQs*, and the expression of most *SiVQ* genes was induced after hormone treatment, suggesting that the expression of *SiVQ* genes might be influenced by hormones. Overall, more cis-acting elements responsive to ABA, SA, and MeJA were found in the promoters of *SiVQs* than those responsive to GA. The expression levels of *SiVQs* were also more strongly upregulated after ABA, SA, and MeJA treatment than after GA treatment. The cis-acting elements of the corresponding hormones were not found in the promoter region of some *SiVQs*, such as *SiVQ03* and *SiVQ20*, but nonetheless, their expression levels were upregulated after hormone treatment. This finding might be caused by the possible presence of cis-acting elements of more than 2000 bp in their promoters, or it might be that their expression was not directly regulated by hormone contents but was influenced by hormone signaling. Previous studies have demonstrated that plant hormones act as signaling compounds in regulating responses to abiotic stresses and thereby affect plant growth and survival [56]. In this study, *SiVQ05*, *SiVQ12*, *SiVQ18*, and *SiVQ28* were significantly upregulated under drought treatment or after treatment with ABA, GA, and SA hormones. We also found that *SiVQ05*, *SiVQ12*, *SiVQ18*, *SiVQ19*, and *SiVQ22* were significantly induced by salt stress or treatment with ABA, GA, and SA hormones, while *SiVQ2a5* showed significant upregulation under cold stress or treatment with ABA and GA hormones, suggesting that these genes might be involved in responding to abiotic stresses through hormone signaling pathways. These results will help to select candidate genes for the subsequent in-depth studies of the role of *VQ* in plant abiotic stress responses.

VQs can interact with WRKY TFs and participate in various plant biological processes [4,57]. In this study, we predicted the potential interaction between SiVQs and SiWRKYs. Interestingly, SiVQ03 was predicted to interact with SiWRKY33, and both were highly expressed in the root, implying that they might be involved in root development. In addition, the W box element (the WRKY binding site) was observed in the promoters of most *SiVQs*, indicating that WRKYs might regulate not only the encoded protein action of *SiVQ* genes but also their gene expression. Further verification and experimental analysis are required to study the physical interactions between SiVQ proteins and SiWRKY TFs in foxtail millet.

## 5. Conclusions

In this research, 32 *SiVQ* genes in foxtail millet were identified and systematically analyzed for the first time. The *SiVQ* genes were classified into seven groups through phylogenetic analysis, and the gene family was shown to have been expanded via chromosome segment duplication. The promoters of *SiVQs* were enriched with the cis-acting elements involved in responses to plant growth and development, abiotic and biotic stresses, and hormone responses. Further expression analysis showed that the expression levels of *SiVQ* genes were induced by abiotic stress and hormone treatments. Protein interaction analysis revealed that 15 SiVQ proteins interacted with SiWRKY TFs. Overall, these results provide a basis for exploring the functions of *VQs* in growth and development in foxtail millet and, more broadly, in plants.

## Figures and Tables

**Figure 1 genes-14-01032-f001:**
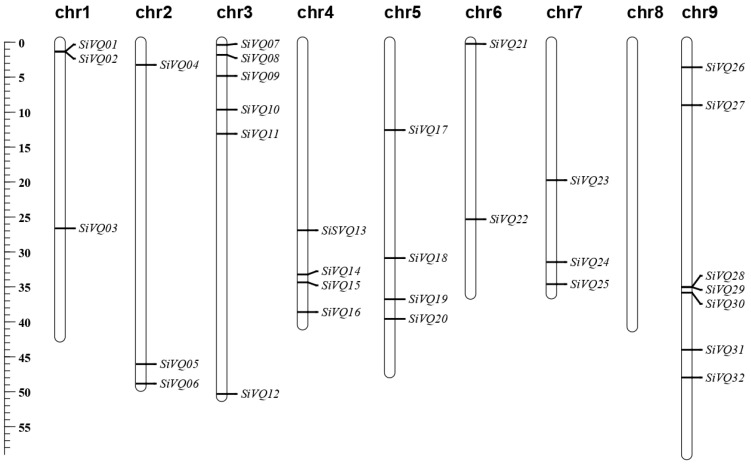
Chromosomal localization of the foxtail millet *SiVQ* genes. The length of the chromosome is displayed on the left using a megabase (Mb) scale.

**Figure 2 genes-14-01032-f002:**
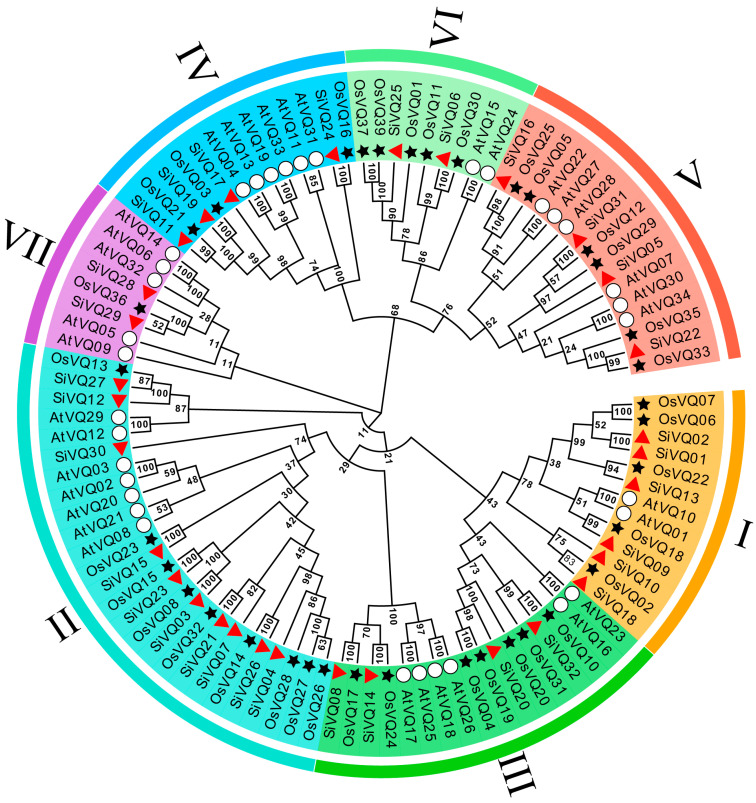
Phylogenetic analysis of VQs from Arabidopsis, rice, and foxtail millet. Numbers I–VII indicate different groups, and the different colors in the outermost circle represent different groups. Hollow circles, black stars, and red triangles represent VQs from Arabidopsis, rice, and foxtail millet, respectively.

**Figure 3 genes-14-01032-f003:**
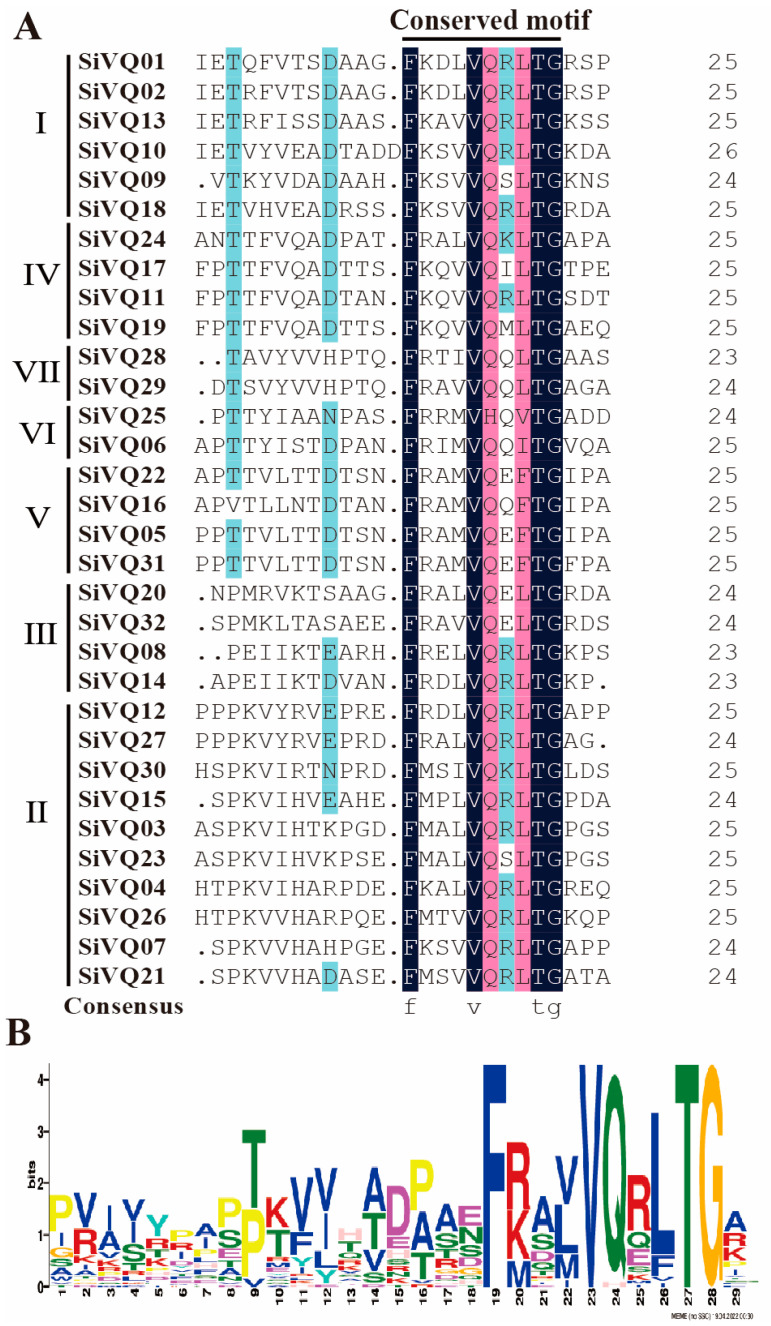
Multiple sequence alignment of foxtail millet VQs: (**A**) multiple sequence alignment of the VQ domain of 32 foxtail millet VQs, with conserved amino acids shaded in different colors; (**B**) conserved motif of SiVQs.

**Figure 4 genes-14-01032-f004:**
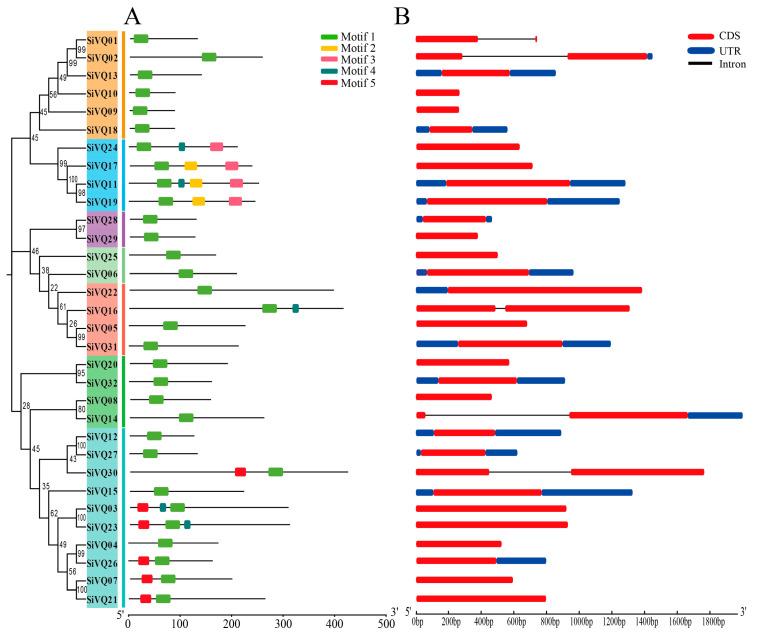
Conserved motifs and gene structure analyses of SiVQs based on phylogenetic relationships in foxtail millet: (**A**) motif composition of foxtail millet VQs, with motifs 1–5 represented by different colored boxes; (**B**) exon–intron structure analyses of *VQ* genes in foxtail millet. red boxes, blue boxes, and black lines represent the exons, 5′ and 3′ untranslated regions, and introns, respectively.

**Figure 5 genes-14-01032-f005:**
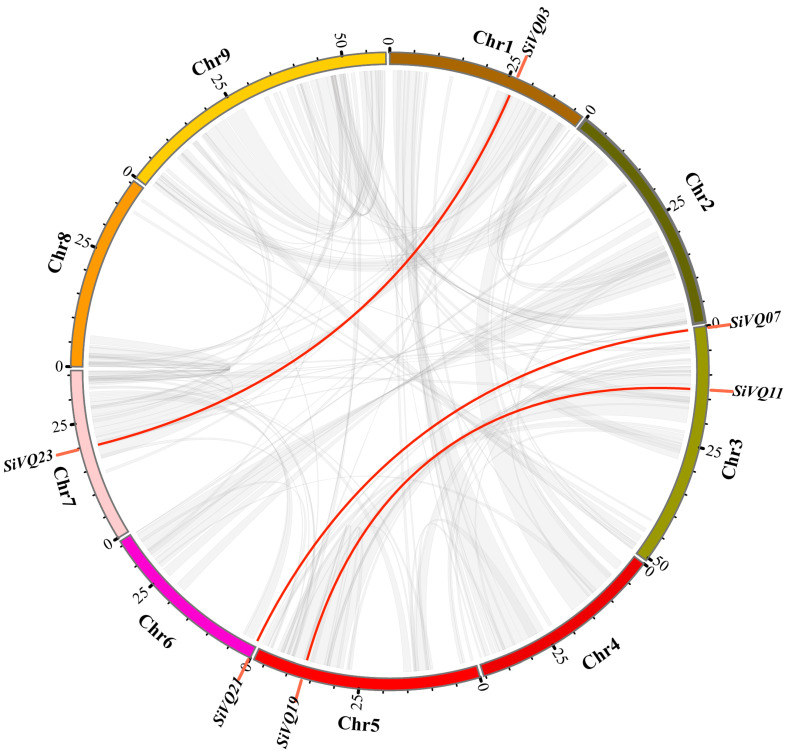
Syntenic analysis of the *SiVQ* genes. The grey lines in the background represent the collinear blocks within the foxtail millet genome, while the red lines highlight the syntenic *SiVQ* gene pairs.

**Figure 6 genes-14-01032-f006:**
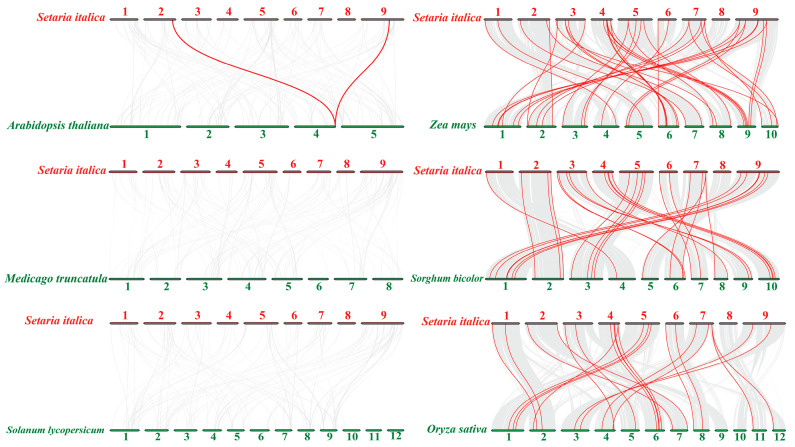
Synteny analysis of *VQ* genes between foxtail millet and six different plant species. The grey lines in the background represent the collinear blocks between the foxtail millet and other plant genomes, while the red lines highlight the collinear *VQ* gene pairs. The numbers represented chromosomes.

**Figure 7 genes-14-01032-f007:**
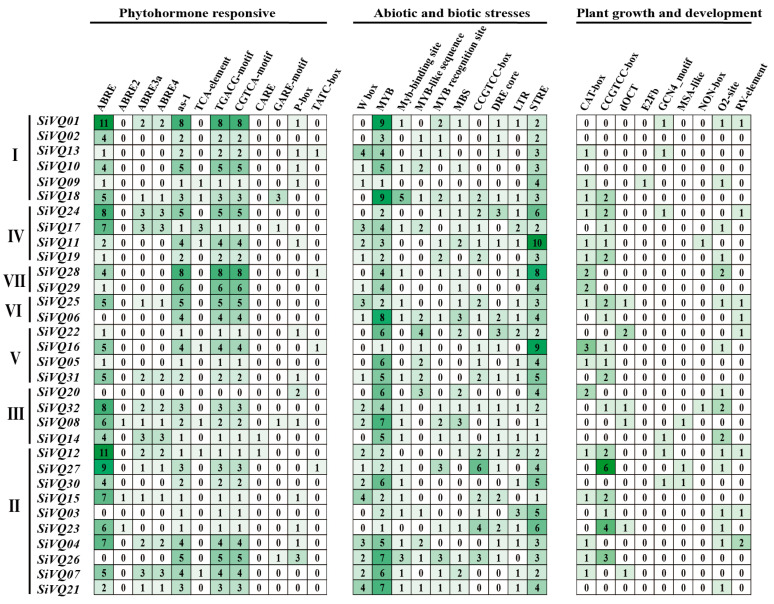
Cis-acting element analysis of the foxtail millet *SiVQ* gene family. Numbers I–VII indicated different groups classified based on phylogenetic relationships.

**Figure 8 genes-14-01032-f008:**
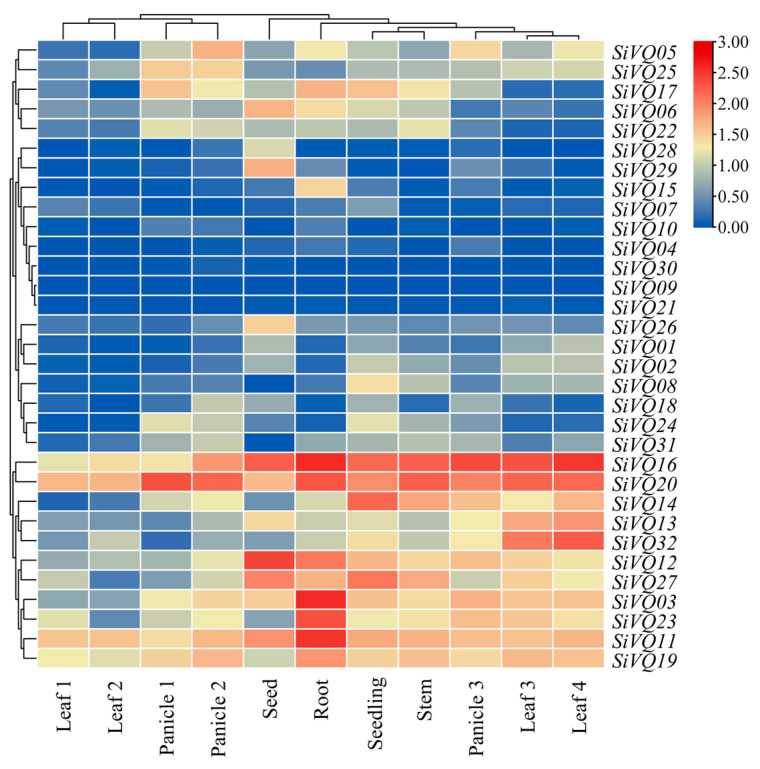
The expression patterns of *SiVQ* genes in various tissues of foxtail millet. Panicle1: panicle of two days after heading; Panicle2: panicle at pollination stage; Panicle3: panicle at filling stage; Leaf1: the uppermost leaf of two weeks; Leaf2: the uppermost second leaf of 30 days; Leaf3: flag leaf at filling stage; Leaf4: the uppermost fourth leaf at filling stage.

**Figure 9 genes-14-01032-f009:**
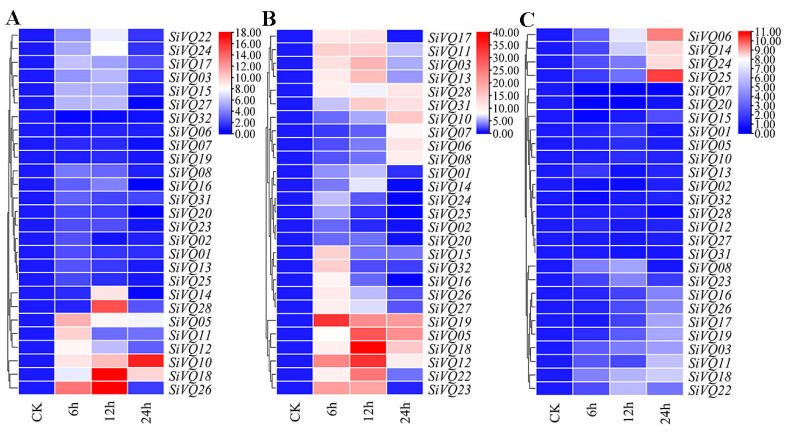
Expression profile of *SiVQ* genes under abiotic stress based on qRT-PCR: (**A**) PEG 6000; (**B**) NaCl, (**C**) 4 °C.

**Figure 10 genes-14-01032-f010:**
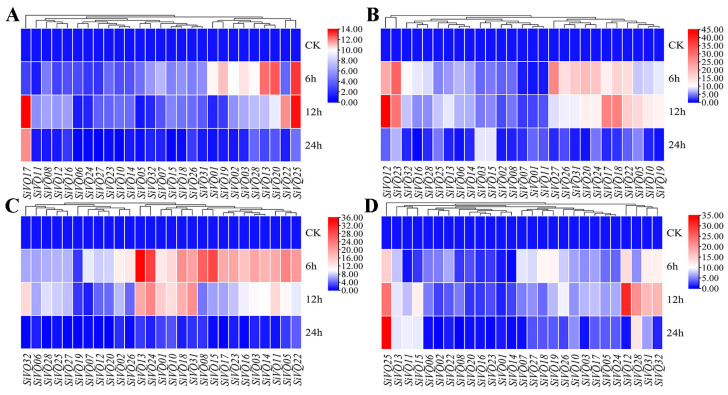
Expression profiles of *SiVQ* genes under hormone treatments based on qRT-PCR: (**A**) gibberellin; (**B**) methyl jasmonate; (**C**) salicylic acid; (**D**) abscisic acid.

**Figure 11 genes-14-01032-f011:**
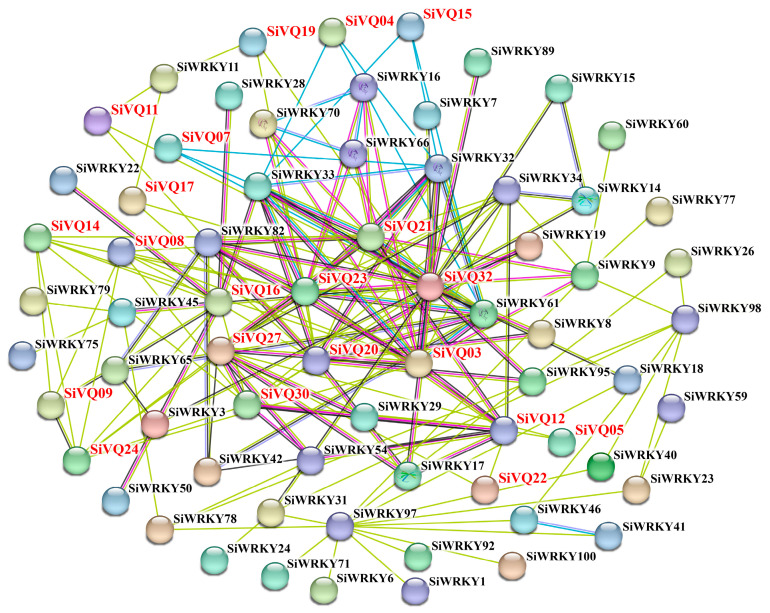
Interaction network between SiVQ proteins and SiWRKY transcription factors. Red and black letters represent SiVQ proteins and WRKY TFs, respectively. Purple and light blue lines represent known interactions determined from the experiments and database, respectively. Red, green, and blue lines represent predicted interactions from gene fusion, proximity, and symbiosis, respectively. Black, light green, and grey lines represent other interactions from coexpression, text mining, and protein homology, respectively. Empty nodes and filled nodes indicate proteins with unknown and known or predicted three-dimensional structures, respectively.

**Table 1 genes-14-01032-t001:** Information on the *SiVQ* gene family in foxtail millet.

Gene ID	Alias	Chr	Position	Protein Length (aa)	Type	Pi ^a^	MW (kD) ^b^	Subcellular Localization ^c^
*SETIT_019438mg*	*SiVQ01*	I	1,341,861−1,342,593	128	LTG	7.71	13.49	mito:11, chlo:1, nucl:1
*SETIT_018158mg*	*SiVQ02*	I	1,345,238−1,346,666	253	LTG	10.05	26.07	chlo:10, nucl:3
*SETIT_019357mg*	*SiVQ03*	I	26,627,391−26,628,299	302	LTG	7.23	31.88	nucl:12.5, nucl_plas:7
*SETIT_033098mg*	*SiVQ04*	II	3,233,035−3,233,550	171	LTG	6.09	17.58	nucl:6, chlo:5, mito:3
*SETIT_033394mg*	*SiVQ05*	II	46,041,085−46,041,753	222	FTG	6.62	22.99	nucl:12, chlo:2
*SETIT_031131mg*	*SiVQ06*	II	48,839,908−48,840,856	204	ITG	6.35	21.21	nucl:13
*SETIT_025350mg*	*SiVQ07*	III	375,390−375,974	220	LTG	10.45	23.79	chlo:6, mito:6, nucl:1
*SETIT_024180mg*	*SiVQ08*	III	1,806,197−1,806,655	157	LTG	5.72	16.52	chlo:6, nucl:4, mito:4
*SETIT_024489mg*	*SiVQ09*	III	4,820,089−4,820,346	85	LTG	5.64	8.95	chlo:8, nucl:2, cyto:2, mito:1
*SETIT_024902mg*	*SiVQ10*	III	9,644,821−9,645,084	87	LTG	5.08	9.22	nucl:7, cyto:5, chlo:2
*SETIT_023066mg*	*SiVQ11*	III	13,094,141−13,095,404	248	LTG	10.46	25.72	chlo:7, nucl:4, mito:3
*SETIT_023655mg*	*SiVQ12*	III	50,301,917−50,302,794	122	LTG	10.45	13.01	mito:7, nucl:3, chlo:2, cyto:2
*SETIT_007480mg*	*SiVQ13*	IV	26,904,107−26,904,951	136	LTG	7.86	14.55	mito:12, nucl:1
*SETIT_007126mg*	*SiVQ14*	IV	33,220,220−33,222,190	256	LTG	5.70	27.14	chlo:10, nucl:3
*SETIT_007257mg*	*SiVQ15*	IV	34,348,031−34,349,338	217	LTG	10.86	22.11	nucl:8, cyto:3, chlo:2
*SETIT_008810mg*	*SiVQ16*	IV	38,605,126−38,606,413	239	FTG	7.33	24.50	nucl:14
*SETIT_004264mg*	*SiVQ17*	V	12,570,630−12,571,331	220	LTG	9.2	22.23	nucl:4.5, nucl_plas:3.5, chlo:3, mito:3, cyto:2
*SETIT_003551mg*	*SiVQ18*	V	30,882,273−30,882,824	85	LTG	6.90	9.11	nucl:6, mito:4, chlo:3
*SETIT_002727mg*	*SiVQ19*	V	36,762,032−36,763,259	241	LTG	10.25	25.22	nucl:14
*SETIT_004625mg*	*SiVQ20*	V	39,570,237−39,570,797	186	LTG	5.36	18.98	nucl:7, mito:5, cyto:2
*SETIT_015774mg*	*SiVQ21*	VI	230,417−231,199	260	LTG	9.57	26.86	nucl:11.5, cyto_nucl:6.5, chlo:1
*SETIT_013900mg*	*SiVQ22*	VI	25,319,524−25,320,889	390	FTG	6.62	38.97	nucl:13
*SETIT_010684mg*	*SiVQ23*	VII	19,742,260−19,743,177	305	LTG	6.43	31.60	chlo:7, mito: 4, nucl:3
*SETIT_011801mg*	*SiVQ24*	VII	31,441,450−31,442,073	207	LTG	9.30	22.31	nucl:11, chlo:1, mito:1
*SETIT_012398mg*	*SiVQ25*	VII	34,606,788−34,607,282	165	VTG	11.21	17.14	chlo:10, nucl:3
*SETIT_037845mg*	*SiVQ26*	IX	3,581,772−3,582,555	160	LTG	10.01	16.48	nucl:8, nucl_plas:5.5, mito:3, chlo:2
*SETIT_038017mg*	*SiVQ27*	IX	9,013,531−9,014,140	129	LTG	9.80	13.09	mito:8, nucl:5
*SETIT_038033mg*	*SiVQ28*	IX	35,017,910−35,018,366	126	LTG	9.68	13.06	chlo:8, nucl:6
*SETIT_038923mg*	*SiVQ29*	IX	35,040,070−35,040,444	124	LTG	7.91	13.01	nucl:7, mito:5, chlo:2
*SETIT_039726mg*	*SiVQ30*	IX	35,814,363−35,816,103	238	LTG	9.10	23.84	chlo:11, nucl:2
*SETIT_036960mg*	*SiVQ31*	IX	44,006,470−44,007,643	209	FTG	9.51	20.88	nucl:8, chlo:5
*SETIT_037859mg*	*SiVQ32*	IX	47,969,328−47,970,225	157	LTG	7.96	16.37	nucl:8, chlo:3, pero:3

Abbreviations: ^a^ pI, theoretical isoelectric point; ^b^ MW, theoretical molecular weight, kDa; ^c^ mito, mitochondria; chlo, chloroplast; nucl, nuclear; plas, plasma membrane; cyto, cytoplasmic; nucl_plas, nuclear and plasma membrane; cyto_nucl, nuclear and cytoplasmic; pero, peroxisome.

**Table 2 genes-14-01032-t002:** Analysis of *SiVQ* genes synteny blocks.

Gene 1	Gene 2	*Ka* ^a^	*Ks* ^b^	*Ka/Ks*	T(MYA) ^c^	Group
*SiVQ03*	*SiVQ23*	0.52951	0.412243	1.28446	31.7	II
*SiVQ11*	*SiVQ19*	0.307855	0.311895	0.987048	24.0	IV
*SiVQ07*	*SiVQ21*	0.742654	0.595114	1.24792	45.7	II

^a^ *Ka* non-synonymous substitution. ^b^
*Ks* synonymous substitution. ^c^ The divergence time (T) is given as millions of years ago (MYA).

## Data Availability

The materials and data in this study are available in the Appendix A or from the corresponding author upon reasonable request.

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
