# Peer review of "Genome-Wide Identification and Characterization of the VQ Motif-Containing Gene Family Based on Their Evolution and Expression Analysis under Abiotic Stress and Hormone Treatments in Foxtail Millet (Setaria italica L.)"

_genes, 2023, doi:10.3390/genes14051032_

Round 1

Reviewer 1 Report

The authors have done an complete study related to function and evolution of VQ motif-containing gene family foxtail millet (Setaria italica L.). In my opinion, this manuscript has high potential for publishing. I have some comments:

-          Line 18: Use italic format for showing gene names. Please apply for whole text and figures.

-          Please provide some references for lines 36-42.

-          Results are repeated in discussion. Please remove them. In discussion, key results should be interpreted.

-          I suggest adding it to line 464: However, duplicated genes showed diverse expression profile. It seems that mutations in critical regions including promoter and coding sites have altered the expression of new members.

Author Response

1.Line 18: Use italic format for showing gene names. Please apply for whole text and figures.

[Response] Thank you for your comments. We have revised the inappropriate expression.

2. Please provide some references for lines 36-42.

[Response] Thank you. We have added this corresponding references (Line 36-43).

3.Results are repeated in discussion. Please remove them. In discussion, key results should be interpreted.

[Response] Thank you for your comments. We have deleted the corresponding repeated results in the discussion section.

4. I suggest adding it to line 464: However, duplicated genes showed diverse expression profile. It seems that mutations in critical regions including promoter and coding sites have altered the expression of new members.

[Response] Thank you for your suggestion. We have added this discussion (Line 481-483).

Reviewer 2 Report

Article review

1. EVALUATION OF THE PAPER MANUSCRIPT

Title of the Manuscript:

"Genome-wide identification and characterization of the VQ 2

motif-containing gene family in their evolution and expression 3

analysis under abiotic stress and hormonal treatments in foxtail 4

millet (Setaria italica L.)"

Manuscript number: genes-2298438

The article genes-2298438 has comprehensively identified and characterized the VQ motif-containing proteins in foxtail millet (Setaria italica L.) based on various bioinformatics approaches. Then, expression patterns of the SiVQ genes in various organs/tissues under different treatments, like high salt, drought, and cold, and four different hormone treatments, like abscisic acid, salicylic acid, gibberellin, and methyl jasmone were explored by using RT-qPCR. The article genes-2298438 is suited to the Genes as per aims and scope. However, to improve the quality of the article, some points should be noted:

-  Authors state that 2 Kb sequences upstream to the transcription start site were retrieved for the prediction of cis-regulatory elements. How did the transcription start site get determined? Furthermore, the PlantCARE is out of date because the authors had to manually search for many well-characterized stress-responsive cis-acting elements, particularly nitrate-responsive cis­­­-elements. As a result, the Results and Discussion section related to this part should concentrate on the stress- and hormone-responsive cis-regulatory elements.

- Move the method to estimate the divergence time as in the note of Table 2 to the Method section and cite the suitable references.

- Please, define specifically expressed, highly expressed, and expressed genes in the Mehod related to the re-analysis of transcriptome data.

- Provide the reference gene for qRT-PCR analysis.

- The relative expression analysis in RT-qPCR validation is extremely limited. If possible, the authors should include more details in their methods, such as the student's t-test, *p < 0.05, or how they use T-test on qRT-PCR values.

- If possible, please carry out a GFP assay to determine the subcellular localization of several interest SiVQ proteins. The reviewer is looking forward to seeing this part in the revised manuscript.

- Also, it would be more significant if the authors perform a gain-of-function or loss-of-function assay of one interest SiVQ gene. The reviewer is looking forward to seeing this part in the revised manuscript.

- Please check for grammatical and spelling mistakes as some in the pdf file. Many words should be italicized. Some duplicated words should be noted to check. 

Author Response

1. Authors state that 2 Kb sequences upstream to the transcription start site were retrieved for the prediction of cis-regulatory elements. How did the transcription start site get determined? Furthermore, the PlantCARE is out of date because the authors had to manually search for many well-characterized stress-responsive cis-acting elements, particularly nitrate-responsive cis-elements. As a result, the Results and Discussion section related to this part should concentrate on the stress- and hormone-responsive cis-regulatory elements.

[Response] Very sorry for our misrepresentation in the previous manuscript. We first obtained the physical location of the cDNA and then blasted it to the genomic sequence to obtain the genomic DNA sequences 2000 bp upstream of the cDNA of the SiVQs (Lines 150-153). We have corrected the inappropriate expression in the revised manuscript. In addition, we manually searched for stress-responsive cis-acting elements according your suggestion, and have revised this part in the revised manuscript (Lines 299-318; Table S5).

2.Move the method to estimate the divergence time as in the note of Table 2 to the Method section and cite the suitable references.

[Response] Thank you very much for pointing this. We have revised it (Line 147-148).

3. Please, define specifically expressed, highly expressed, and expressed genes in the Mehod related to the re-analysis of transcriptome data.

[Response] Thank you for your comments. We have added it (Line 159-163).

4. Provide the reference gene for qRT-PCR analysis.

[Response] Thank you for your comments. We have previously listed the internal control genes in Table S1, and we have also added it in the revised manuscript (Line 188-189).

5. The relative expression analysis in RT-qPCR validation is extremely limited. If possible, the authors should include more details in their methods, such as the student's t-test, *p < 0.05, or how they use T-test on qRT-PCR values.

[Response] Thank you for your comments. We have revised it (Line 189-191).

6. If possible, please carry out a GFP assay to determine the subcellular localization of several interest SiVQ proteins. The reviewer is looking forward to seeing this part in the revised manuscript.

[Response] Thank you for your comments. We do not currently have the results of GFP assay to determine the subcellular localization of interest SiVQ proteins, but we have used the subcellular localization prediction tool WoLF PSORT to predict the likely location of the SiVQ proteins in the manuscript. We will perform a GFP assay to determine the subcellular localisation of SiVQ once we have determined its function.

7. Also, it would be more significant if the authors perform a gain-of-function or loss-of-function assay of one interest SiVQ gene. The reviewer is looking forward to seeing this part in the revised manuscript.

[Response] Thank you very much for pointing this out. The study of gene function is very important, but it takes a long time to obtain stable gain-of-function or loss-of-function material, so we do not yet have results for this part of this study. we have discussed several potentially functional SiVQ genes in the manuscript and will select interest SiVQs to further explored its molecular functions and regulatory mechanisms in subsequent studies.

8. Please check for grammatical and spelling mistakes as some in the pdf file. Many words should be italicized. Some duplicated words should be noted to check.

[Response] Thank you for your comments. We have revised font problem for the statistical numbers and inappropriate expression.

Reviewer 3 Report

Reviewed paper showed complex analysis of foxtail millet genome in terms of the VQ motif-containing proteins.

I have one general question - why the analysis was performed for this particular species? Duplication of papers that only deal with another species (than Arabidopsis) is now avoided, thus please precise why for foxtail millet it so important.

Two detailed comments:

1) I suggest to add TF abbreviation for transcriptional factor 

2) please explain why these particular conditions (200 mM NaCl, 20% PEG 6000, 200µM ABA - not other concentrations, etc.) were used as stress factors?

Author Response

1. I have one general question - why the analysis was performed for this particular species? Duplication of papers that only deal with another species (than Arabidopsis) is now avoided, thus please precise why for foxtail millet it so important.

[Response] Thank you very much for pointing this out. Foxtail millet is one of the traditional food crops in China, which has the characteristics of small genome size, short growth period and strong abiotic stress resistance. The potential tolerance to abiotic stress of foxtail millet has made it a research material for the study of molecular mechanisms of abiotic stress resistance (Line 81-88). VQ proteins can affect the transcriptional activity of transcription factors to be involved in plant growth and development, biotic and abiotic stress responses, hormone signaling pathways. Therefore, we used foxtail millet as material to investigate whether VQ genes play an important role in the strong resistance of foxtail millet to abiotic stresses and to provide a basis for subsequent studies on the molecular mechanisms of abiotic stress resistance.

2. Two detailed comments:

1) I suggest to add TF abbreviation for transcriptional factor

[Response] Thank you for your comments. We have revised them.

2) please explain why these particular conditions (200 mM NaCl, 20% PEG 6000, 200µM ABA - not other concentrations, etc.) were used as stress factors?

[Response] Thank you for your comments. We have selected these specific concentrations as stress factors by referring to the treatments in previous published articles, and the corresponding references are added to the manuscript (Line174-177).
